# Gender Differences in Obstructive Sleep Apnea: A Preliminary Clinical and Polysomnographic Investigation

**DOI:** 10.3390/neurolint17060085

**Published:** 2025-05-29

**Authors:** Alessandra Castelnuovo, Sara Marelli, Salvatore Mazzeo, Francesca Casoni, Paola Proserpio, Alessandro Oldani, Alessandro Bombaci, Elisa Bortolin, Giulia Bruschi, Federica Agosta, Massimo Filippi, Luigi Ferini-Strambi, Maria Salsone

**Affiliations:** 1Department of Psychology, Vita-Salute San Raffaele University, 20132 Milan, Italy; castelnuovo.alessandra@hsr.it (A.C.); e.bortolin@studenti.unisr.it (E.B.); g.bruschi3@studenti.unisr.it (G.B.); agosta.federica@hsr.it (F.A.); ferinistrambi.luigi@hsr.it (L.F.-S.); 2Sleep Disorders Center, IRCCS San Raffaele Scientific Institute, 20127 Milan, Italy; marelli.sara@hsr.it (S.M.); casoni.francesca@hsr.it (F.C.); proserpio.paola@hsr.it (P.P.); oldani.alessandro@hsr.it (A.O.); 3Department of Neurology, Vita-Salute San Raffaele University, 20132 Milan, Italy; mazzeo.salvatore@hsr.it (S.M.); bombaci.alessandro@hsr.it (A.B.); filippi.massimo@hsr.it (M.F.); 4IRCCS Istituto Policlinico San Donato, 20097 Milan, Italy; 5Neurology Unit, IRCCS San Raffaele Scientific Institute, 20132 Milan, Italy; 6Neuroimaging Research Unit, Division of Neuroscience, IRCCS San Raffaele Scientific Institute, 20132 Milan, Italy; 7Neurorehabilitation Unit, IRCCS San Raffaele Scientific Institute, 20132 Milan, Italy; 8Neurophysiology Service, IRCCS San Raffaele Scientific Institute, 20132 Milan, Italy

**Keywords:** obstructive sleep apnea (OSA), cognition, gender

## Abstract

Background/Objectives: Gender differences influence the clinical manifestations, progression, and treatment response in obstructive sleep apnea (OSA) syndrome, suggesting the existence of distinct gender-related phenotypes potentially driven by anatomical, physiological, and hormonal factors. However, the impact of gender on OSA-related cognitive profiles remains unknown. This study aimed to investigate the neuropsychological and polysomnographic (PSG) differences between OSA females and males in order to detect the impact of gender on clinical manifestation and PSG features. Methods: Data were collected from 28 OSA patients (14 females and 14 males matched for age, education, and disease severity). All patients performed a complete neuropsychological evaluation, Epworth sleepiness scale, and whole-night PSG. To evaluate the relationship between specific sleep profiles and cognitive performance, PSG parameters were correlated to scores obtained on neuropsychological tests. Results: Both male and female groups performed within the normal range across all administered neuropsychological tests, according to Italian normative values. Compared with OSA males, female patients showed significantly lower values on the Rey–Osterrieth Complex Figure (ROCF) Recall Test. By contrast, no significant statistical clinical difference emerged between the two OSA groups in terms of clinical manifestation and sleep parameters. Conclusions: This study improves the knowledge on gender-related cognitive impairment in OSA patients. Our preliminary findings demonstrate that the ROCF Recall Test may be altered in OSA females, but not in males. Further longitudinal studies are needed to investigate whether OSA female patients will develop a frank dementia over time.

## 1. Introduction

Gender differences may commonly occur in the context of several neurological disorders, especially in sleep disorders such as insomnia [1], restless legs syndrome (RLS) [1,2], rapid eye movement (REM), sleep behavior disorder (RBD) [3], and, in particular, in obstructive sleep apnea (OSA) [1,4]. OSA is a very common sleep-related breathing disorder clinically characterized by repeated episodes of upper airway obstruction during sleep, chronic intermittent hypoxia, and sleep fragmentation [5]. It is more prevalent in men than women, although it increases with age in both genders [6,7]. Furthermore, the difference in prevalence between men and women diminishes after the age of 60, partly due to menopausal status [6].

A growing body of evidence highlights the detrimental effects of OSA on neuropsychological functioning in adults. Meta-analytic findings indicate that, particularly in younger and middle-aged populations, OSA is associated with deficits in various cognitive domains, including memory [8], vigilance, and attention [9,10], as well as executive function and motor coordination [11,12,13]. Although the underlying pathophysiological mechanisms remain incompletely understood, both intermittent hypoxemia and sleep fragmentation are implicated, with hypoxemia potentially exerting a more pronounced adverse effect on cognitive outcomes [14]. As a result of sleep fragmentation, most patients may also exhibit daytime clinical symptoms, including excessive daytime sleepiness (EDS) and morning headaches [15,16]. OSA-related cognitive impairment, however, can be partially reversible after Continuous Positive Airway Pressure (CPAP), the gold standard treatment for OSA [17]. Finally, OSA itself can contribute as an independent risk factor for cardiovascular, metabolic, and psychiatric disorders, thus representing a global research priority [18].

Although women often face greater challenges in accessing proper diagnostic care than men, gender differences in OSA have been detected regarding clinical presentation, polysomnographic features, functional status, comorbidities, and therapeutic management [19,20]. Notably, OSA females may present a later onset, with disease severity increasing only after the age of 50 years and depending, in part, on hormonal patterns and the menopause state [15,16,21]. Likewise, the onset of menopause in women has been associated with an elevated risk of developing Alzheimer’s disease (AD), potentially due to factors such as estrogen deficiency, hypertension, enhanced endothelial dysfunction, and systemic inflammation [22]. These mechanisms are also activated by OSA, suggesting that OSA and AD may share intricate pathophysiological pathways. However, sex-specific differences in OSA prevalence and its clinical manifestations remain unclear [23]. Snoring is often under-observed in females as they are less likely to undergo a clinical assessment with their bed partner [24]. Finally, OSA females frequently exhibit other neurological symptoms related to the disease, such as morning headache, depression, a higher number of awakenings, and nocturia, compared to males [25]. Women affected by OSA more commonly report symptoms of insomnia compared to men, which is likely attributable to longer-lasting partial upper airway obstructions and apnea–hypopnea episodes occurring during REM sleep, leading to repeated arousals and increased sleep fragmentation [26,27]. Consequently, in women, the pronounced sleep fragmentation caused by OSA may exacerbate the dysfunction of the glymphatic system, thereby heightening the risk of neurodegenerative processes [28]. Such findings support the hypothesis of two distinct gender-related OSA phenotypes, which may be partly explained by different anatomical and physiological characteristics. Four physiological traits have been identified as contributors to OSA pathogenesis: upper airway anatomy, upper airway gain (tone/responsiveness of the upper airway muscles), arousal threshold (arousal in response to a respiratory event), and loop gain (relative stability of the respiratory control system) [6,29]. Moreover, decreased lung volume during sleep and the potential nocturnal displacement of rostral fluids from the legs to the neck area may further decrease the longitudinal traction of the pharynx and narrow the pharyngeal lumen, increasing its collapsibility [6,29]. Won and colleagues found that women have a lower loop gain, less airway collapsibility, and a lower arousal threshold in non-rapid eye movement (NREM) sleep [26]. Individual OSA phenotypes may be explained by one or more endotypes that underlie the disease mechanism [6].

Little is known regarding the relationship between gender differences and cognitive impairment in OSA. Thus, the primary aim of the present study was to investigate the impact of gender on cognitive functioning in patients with OSA.

## 2. Materials and Methods

### 2.1. Participants

In our study, a total of 28 OSA patients (14 females and 14 males matched for age, education, and disease severity) were recruited from the Sleep Disorder Center, IRCCS San Raffaele Hospital, Milan. All participants were right-handed, monolingual Italian native speakers, and had normal or corrected-to-normal visual acuity. All female participants were postmenopausal and did not receive hormone therapy. Inclusion criteria were as follows: (i) diagnosis of moderate or severe OSA (apnea/hypopnea index [AHI] > 15); (ii) no symptoms of cognitive deterioration (Mini-Mental State score > 24). Additionally, participants had no evidence of stroke, neurological disorder, major psychiatric disorder, uncontrolled hypertension, or respiratory failure and had no current use of any psychoactive medications. All participants signed an informed consent form for their data to be stored for further research. All participants provided their written informed consent to the experimental procedure, which was previously approved by the local ethical committee CET 94-2023.

### 2.2. Sleep Apnea Assessment

All patients underwent a full night of polysomnography (PSG) according to our previous protocol [30]. The following signals were recorded: electroencephalogram (six channels, including C3 or C4 and O1 or O2, referred to as the contralateral mastoid); electro-oculogram; electromyography (EMG) of the submentalis muscle; EMG of the right and left tibialis anterior muscles; electrocardiogram (ECG; one derivation), according to the American Academy of Sleep Medicine scoring criteria [30]. We monitored the sleep respiratory pattern using oral and nasal airflow thermistors and/or nasal pressure cannula, thoracic and abdominal respiratory effort strain gauge, and by monitoring oxygen saturation (pulse oximetry). Two physicians (FC and AO) with experience in sleep medicine scored the respiratory events according to the American Academy of Sleep Medicine scoring criteria [30].

### 2.3. Cognitive Measures

An accurate and complete neuropsychological evaluation has been performed in all OSA patients including the assessments in the following areas (see Appendix A for details): (1) global functioning (Mini-Mental State Examination-MMSE) [31]; (2) memory: short-term verbal and spatial memory (Digit Span Forward and Corsi Block-Tapping Test) [32,33,34], long-term verbal memory (Rey Auditory Verbal Learning Test: learning, recall, and recognition) [35], and working memory (Digit Span Backward) [36]; (3) attention and executive functions: nonverbal reasoning (Raven’s Progressive Matrices) [33], visual selective attention (Attentive Matrices) [33], visuo-constructional abilities (Rey–Osterrieth Complex Figure) [37]; (4) language: verbal fluency (Phonemic and Semantic Cues) [38,39], and linguistic comprehension (Token Test) [40]. The administration of the neuropsychological test battery lasted approximately 60 min.

### 2.4. Statistical Analysis

Patient groups were characterized using means and standard deviations (SD). We tested for normality in the distribution of the data using the Kolmogorov–Smirnov test. Depending on the distribution of the data, we used *t*-tests or non-parametric Mann–Whitney U Tests for between-group comparisons, and Spearman’s correlation was used to assess the relationship between clinical, polysomnographic data, and scores obtained on neuropsychological evaluation. When comparing OSA female and OSA male groups, the non-parametric Mann–Whitney U Test was used to assess differences in clinical–demographic, polysomnographic variables, and neuropsychological scores. Missing data were handled using multiple imputations. Statistical analysis was performed using SPSS program version 21.0.

## 3. Results

### 3.1. Demographic and Clinical Features

OSA female and male groups showed the same proportion of patients with moderate (*n* = 9) and severe OSA (*n* = 5). Table 1 summarizes the main characteristics of the total OSA group, female and male. There were no statistically significant differences between groups in terms of age, weight, Body Mass Index (BMI), and disease duration. Educational levels, which are known to possibly influence cognitive performance [41,42], were not statistically different between the two groups. The percentage of men suffering from gastroesophageal reflux disease was statistically higher compared to women (*p* = 0.025). There was a trend of an increased percentage of OSA females suffering from hypertension. Regarding the sleep profile, a slight but not significant difference emerged when comparing the two OSA groups (Table 1). OSA females showed a lower value of the oxygen desaturation index (ODI), sleep latency (SL), N2%, and REM latency, with an increased value of the periodic limb movement (PLM) index and the number of awakenings as compared to OSA males (Table 2, Figure 1A). The scoring of respiratory events did not show differences between the two physicians, confirming a high level of inter-rater reliability.

### 3.2. Neuropsychological Performances

Both groups performed within the normal range across all administered neuropsychological tests, according to Italian normative values. The neuropsychological evaluation revealed that OSA females had significantly lower scores than OSA males on the Rey–Osterrieth Complex Figure Recall Test (*p* = 0.027) (Table 3, Figure 1B). No other tests showed statistically significant differences between groups (Table 3).

## 4. Discussion

The primary aim of the present study was to examine the impact of gender on cognitive performance in OSA patients. Our findings revealed that female OSA patients scored significantly lower on the Rey–Osterrieth Complex Figure (ROCF) Recall Test than male OSA patients.

The recent literature has established that OSA causes cognitive dysfunction, particularly affecting memory, attention, and executive abilities [11,43,44,45]. Specifically, OSA has been linked to difficulties in concentration [46], memory [47,48], learning new tasks [49], emotional regulation, and motivation [48]. Furthermore, patients also report greater challenges in maintaining attention and vigilance [48]. Recent research has also suggested that the impact of OSA on cognitive performance may vary as a function of gender [50]. Gender differences in cognitive performance have been widely described. Specifically, females typically outperform males on tasks involving perceptual speed, verbal, and fine motor skills [51,52,53], while males often excel in mental rotation, navigation tasks, and spatial orientation [51,54,55,56]. In the context of AD, gender is a significant risk factor, with women representing approximately two-thirds of diagnosed cases [57]. Various biological, genetic, and socio-cultural factors may contribute to this discrepancy. For instance, estrogen is believed to have a neuroprotective effect against hypermetabolism and reduced neuronal mitochondrial function in the hippocampus and prefrontal cortex, regions involved in learning and memory decline during menopause and the early stages of AD [58,59]. In women, the precipitous decrease in the levels of estrogen at menopause has been reported to affect both sleep architecture and memory consolidation [59], potentially accelerating cognitive decline and increasing the risk of developing AD [60]. It can be speculated that OSA may further increase this risk, potentially accelerating the shift to neurodegenerative processes. This could be especially important considering that OSA is often underdiagnosed in women [20], suggesting that its role in long-term cognitive decline in women should be further explored. Accordingly, in our cohort, the female group had a mean age of 64 years, raising the possibility that menopausal status may have influenced the observed differences in cognitive performance, notably in the memory domain, compared to the male group.

The significantly worse performance of the female OSA group in the ROCF Recall Test is not surprising. The ROCF Recall Test is a comprehensive neuropsychological tool that evaluates not only visual memory but also multiple cognitive domains, including spatial organization, attention, planning, and executive functioning. It is widely used to study the neural correlations of brain function in healthy individuals as well as in patients with dementia and other brain disorders. Notably, previous research has shown that ROCF Recall Test scores are associated with AD biomarkers. For instance, Park et al. [61] identified the ROCF Recall Test as the only neuropsychological variable correlated with amyloid-beta (Aβ) burden in patients with mild cognitive impairment (MCI). Our findings are particularly relevant in light of the increasing evidence linking OSA to neurodegeneration, particularly AD. Sleep disturbances, such as those caused by OSA, are thought to contribute to AD pathology through mechanisms including sleep fragmentation, the disruption of slow-wave sleep (N3), glymphatic dysfunction, intermittent hypoxia, and increased intracranial pressure due to apneic events [62]. OSA has been associated with an elevated risk of developing AD [63], earlier onset MCI or AD [64], and worsening cognitive and functional states in advanced stages of AD, with greater AHI severity correlating with poorer outcomes. These findings support the notion that OSA may function as a “silent” contributor to brain damage [28,65,66,67]. On the other hand, CPAP therapy has demonstrated the potential to delay cognitive decline in OSA patients [64].

Our study has several limitations. First, the small sample size limits the generalizability, reliability, and interpretability of our findings, which may not be representative of the broader OSA population. However, the limited sample size, especially among women, is not unexpected, as it is well reported that OSA is significantly more prevalent in men compared to women. Indeed, in the range of 40–60 years and at the AHI/RDI level ≥ 30, the prevalence of OSA has been estimated at 15.2% among males and only 2.6% among females [21,68]. Given the exploratory nature of this study, multiple comparison corrections were not applied, increasing the risk of type I errors. Second, the absence of a control group prevents us from comparing cognitive performance between individuals with and without OSA, limiting our ability to determine whether the observed cognitive patterns are attributable to the sleep disorder or reflect normal gender-related variability. Further, estrogen levels were not assessed among female participants, despite their potential as confounding factors in cognitive performance. Lastly, we lacked follow-up data to evaluate the long-term cognitive trajectories of our participants and investigate the potential changes in gender-related differences over time. Despite these limitations, our study has a significant strength: the inclusion of OSA patients who were extensively characterized from clinical, neuropsychological, and sleep parameter perspectives. To the best of our knowledge, this is one of the few studies specifically examining the effects of sex on cognitive performance in OSA patients.

In conclusion, our findings highlight the importance of considering gender-specific differences when assessing cognitive function in patients with OSA, supporting the need for earlier and more targeted cognitive screenings in women. This could potentially improve the clinical evaluation and management of these patients, paving the way for a new line of tailored treatment strategies to prevent more severe manifestations of cognitive decline. Future longitudinal research is needed to track cognitive trajectories in OSA patients and explore how gender-related differences evolve, clarifying whether the lower visuospatial memory scores observed in OSA females may reflect a higher vulnerability to cognitive decline or normal gender variability. Expanding the sample size, including a control group, and evaluating sleep microstructure along with additional confounders (including estrogen levels and AHI subtypes) will provide further insights into the interplay between sex, OSA, and cognitive function.

## Figures and Tables

**Figure 1 neurolint-17-00085-f001:**
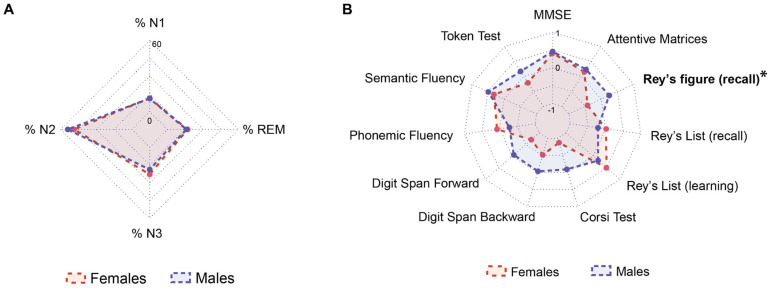
(**A**) Sleep profiles in female (red) and male (blue) patients. Values are expressed as percentages. (**B**) Neuropsychological profiles in female (red) and male (blue) patients. Values are expressed as z-scores, based on mean and standard deviation reported in normative studies for each test. Bold characters indicate statistically significant differences between cohorts: * *p* < 0.05.

**Table 1 neurolint-17-00085-t001:** Demographic and clinical variables of OSA patients in the total group, female group, and male group.

Variables	Total Group (*n* = 28)	Female (*n* = 14)	Male (*n* = 14)	*p*-Value
Demographics				
Age (years)	64.82 ± 11.20	64.50 ± 10.83	65.14 ± 11.95	0.765 ^a^
Education (years)	12.03 ± 4.29	12.00 ± 4.94	12.07 ± 3.71	0.925 ^a^
Weight (kg)	91.53 ± 16.27	82.00 ± 15.67	96.30 ± 15.06	0.086 ^a^
Height (m)	1.67 ± 0.12	1.54 ± 0.07	1.74 ± 0.07	0.005 ^a^
BMI (kg/m^2^)	32.32 ± 4.16	33.25 ± 4.57	31.67 ± 3.97	0.626 ^a^
Duration of disease (years)	5.08 ± 3.18	5.00 ± 3.79	5.14 ± 2.71	0.677 ^a^
Familiarity for dementia (%)	7.4	7.1	7.7	0.957 ^b^
Concomitant Pathologies				
Hypertension (%)	48.1	64.3	30.8	0.082 ^b^
Heart disease (%)	25.9	35.7	15.4	0.228 ^b^
COPD/asthma (%)	7.4	14.3	0	0.157 ^b^
Diabetes (%)	11.1	14.3	7.7	0.586 ^b^
Hypercholesterolemia (%)	37.0	42.9	30.8	0.516 ^b^
Gastroesophageal reflux (%)	14.8	0	30.8	0.025 ^b^
Psychiatric disorders (%)	29.6	28.6	30.8	0.901 ^b^
Others (%)	25.9	28.6	23.1	0.745 ^b^

Results are expressed using mean ± standard deviation. ^a^ = Mann–Whitney U Test; ^b^ = X^2^ test; BMI: Body Mass Index; COPD: chronic obstructive pulmonary disease.

**Table 2 neurolint-17-00085-t002:** Sleep profiles of OSA patients in the total group, female group, and male group.

Variables	Total Group (*n* = 28)	Female (*n* = 14)	Male (*n* = 14)	*p*-Value
Respiratory profile				
Apnea Hypopnea Index (AHI)	34.07 ± 23.62	34.42 ± 28.47	33.75 ± 19.32	0.550 ^a^
Oxygen Desaturation Index (ODI)	30.85 ± 14.85	22.47 ± 11.78	35.88 ± 14.69	0.065 ^a^
Apnea Hypopnea Index Supine (AHI supine)	37.68 ± 27.91	37.34 ± 31.06	38.24 ± 23.91	0.612 ^a^
Sleep Macrostructure				
Total Sleep Time (TST—min)	402.82 ± 60.28	412.17 ± 68.73	391.60 ± 53.77	0.361 ^a^
Sleep Latency (SL—min)	11.73 ± 5.59	9.50 ± 5.09	14.40 ± 5.41	0.079 ^a^
Wake After Sleep Onset (WASO—min)	58.82 ± 30.40	61.83 ± 35.80	55.20 ± 26.04	1 ^a^
Sleep Efficiency (%SE)	83.84 ± 8.17	84.75 ± 8.48	82.76 ± 8.63	0.584 ^a^
Number of Awakenings	15.27 ± 10.62	16.33 ± 11.16	14.00 ± 11.07	0.647 ^a^
% N1	11.25 ± 8.30	11.37 ± 7.29	11.12 ± 10.28	0.715 ^a^
% N2	51.13 ± 11.99	48.72 ± 2.75	54.02 ± 18.20	0.201 ^a^
% N3	21.09 ± 13.01	22.93 ± 12.43	18.88 ± 14.80	0.584 ^a^
% REM	16.53 ± 8.64	16.98 ± 8.09	15.98 ± 10.22	0.855 ^a^
REM Latency (min)	126.68 ± 55.72	115.58 ± 44.28	140.00 ± 70.02	0.583 ^a^
Periodic Leg Movement Index (PLM Index)	14.09 ± 22.23	19.15 ± 27.95	8.02 ± 13.20	0.0848 ^a^

Results are expressed using mean ± standard deviation. ^a^ = Mann–Whitney U Test; N1: Sleep Stage 1; N2: Sleep Stage 2; N3: slow-wave sleep; REM: rapid eye movement; PLMs: periodic limb movements.

**Table 3 neurolint-17-00085-t003:** Neuropsychological profiles of OSA patients in the total group, female group, and male group.

Variables	Total Group (*n*= 28)	Female (*n* = 14)	Male (*n* = 14)	*p*-Value
Mini-Mental State Evaluation (MMSE)	28.11 ± 2.11	28.00 ± 2.74	28.21 ± 1.31	0.391 ^a^
Token Test	32.61 ± 3.99	32.00 ± 5.03	33.23 ± 2.65	0.835 ^a^
Semantic Fluency	42.18 ± 10.33	41.43 ± 8.09	42.93 ± 12.45	0.927 ^a^
Phonemic Fluency	31.53 ± 12.07	33.36 ± 12.41	29.71 ± 11.89	0.395 ^a^
Digit Span Forward	5.43 ± 1.29	5.07 ± 1.07	5.78 ± 1.42	0.195 ^a^
Digit Span Backward	4.14 ± 1.38	3.86 ± 1.03	4.43 ± 1.65	0.475 ^a^
Corsi Test	4.86 ± 1.11	4.43 ± 1.02	5.28 ± 1.07	0.074 ^a^
Rey’s List (learning)	42.18 ± 10.44	43.57 ± 9.32	40.78 ± 11.64	0.382 ^a^
Rey’s List (recall)	8.46 ± 2.97	8.78 ± 2.72	8.14 ± 3.28	0.488 ^a^
Rey’s List (recognition)	13.61 ± 1.40	13.93 ± 1.21	13.28 ± 1.54	0.233 ^a^
False Positive at Rey’s List	1.03 ± 1.37	0.78 ± 0.97	1.28 ± 1.68	0.621 ^a^
Rey’s Figure (recall)	16.11 ± 6.55	13.57 ± 6.22	18.64 ± 6.06	0.027 ^a^
Raven Matrices	28.07 ± 5.66	27.38 ± 6.56	28.71 ± 4.84	0.752 ^a^
Attentive Matrices	48.25 ± 10.71	46.86 ± 12.54	49.64 ± 8.75	0.908 ^a^
Rey’s Figure (copy)	32.00 ± 4.08	31.11 ± 4.67	32.89 ± 3.32	0.138 ^a^

Results are expressed using mean ± standard deviation. ^a^ = Mann–Whitney U Test; Italian population normal values: MMSE: >23.80; Token Test: >26.5; verbal fluency with semantic cues: >25; verbal fluency with phonemic cues: >17; Digit Span Forward: >4.26; Digit Span Backward: >2.65; Corsi Block-Tapping Test: >3.46; Rey Auditory Verbal Learning Test (learning): >28.53; Rey Auditory Verbal Learning Test (recall): >4.69; Rey Auditory Verbal Learning Test (recognition): >8; Rey–Osterrieth Complex Figure (recall): >9.47; Raven’s Progressive Matrices: >18; Attentive Matrices: >31; Rey–Osterrieth Complex Figure (copy): >28.88.

## Data Availability

Data that support the findings of this study will be shared upon request from any qualified investigator.

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
