# Peer review of "Gender Differences in Obstructive Sleep Apnea: A Preliminary Clinical and Polysomnographic Investigation"

_2035-8377, 2025, doi:10.3390/neurolint17060085_

Round 1

Reviewer 1 Report

Comments and Suggestions for Authors

Introduction:
The statement that OSA is more prevalent in men than in women should be revised to clarify that this is particularly true during the premenopausal period. In the postmenopausal period, the prevalence of OSA becomes similar between genders.

Sample size:
The number of patients included in the study is too small to allow for generalizable or meaningful conclusions. The limited sample size substantially weakens the reliability and interpretability of the results.

Methods:
Further details regarding the tests used in the study should be provided to allow proper evaluation of the methodology.

Sociodemographic characteristics:
The authors should report the educational level of the participants, as this could influence neurocognitive performance.

Sleep stage terminology:

  • The authors should use the current standard terminology (N1–N3) instead of the outdated S1–S3 classification.

  • The term “SWS” should be avoided in favor of “N3,” in accordance with AASM guidelines.

Results:
The statement “although neuropsychological tests were in the range of Italian normal value for both groups” contradicts the subsequent assertion that women with OSA had lower scores. This comparison appears statistically inappropriate and should be clarified.

Reviewer 2 Report

Comments and Suggestions for Authors

The manuscript investigates gender differences in cognitive profiles among OSA patients, focusing on neuropsychological and polysomnographic (PSG) measures.

Overall, the study addresses an important gap in OSA research which has remained understudied.

However, there are many caveats which need to be addressed before the manuscript is considered further.

I provide detailed comments and constructive suggestions for improvement.

  1. There’s a major conceptual problem in the study which needs serious consideration. Without healthy controls, it’s unclear if differences are OSA-specific or reflect normal gender variability.
  2. Menopausal status or estrogen levels could confound results but weren’t measured.
  3. EEG spectral analysis (e.g., slow-wave activity) might better explain cognitive differences than macroarchitecture.
  4. The abstract is not clear. Why should gender differences in OSA-related cognitive impairment be expected? Explain this
  5. The phrase "Italian normal range" is not clear. Does this refer to normative data or clinically significant impairment?
  6. The correlation between NREM sleep 2 and semantic fluency is mentioned but not contextualized (e.g., effect size, clinical relevance).
  7. The logic in the introducvtion for focusing on visuospatial memory (Rey-Osterrieth test) is clearly explained.. Why not other domains like executive function or attention?
  8. The link between OSA, gender, and cognitive decline is assumed rather than justified as no mechanistic explanation (e.g., hormonal influence, sleep fragmentation) is provided.
  9. The introduction relies too much on prevalence studies (e.g., OSA being more common in men) without linking to cognitive outcomes.
  10. The methods has some caveats. While age, education, and disease severity were matched, key confounders like menopausal status, hormone therapy, or AHI subtypes were ignored.
  11. There’s no mention of inter-rater reliability for scoring respiratory events or sleep stages.
  12. No control for practice effects or test-retest reliability, which is critical given the subtle differences reported.
  13. Many results (e.g., Token Test, semantic fluency) are highlighted despite lacking statistical significance (Table 3).
  14. With 15+ neuropsychological tests, the risk of Type I error is high. No adjustment (e.g., Bonferroni) was applied.
  15. There’s no details on missing values or how they were handled (e.g., for PSG parameters).
  16. The NREM2-semantic fluency correlation is reported without effect size or scatterplot, making it hard to evaluate.
  17. Table 3: in p values, commas are used instead of dots (.) This makes it confusing with commas used in other cells.
  18. The Rey-Osterrieth difference is mentioned as "subclinical cognitive impairment," but scores were within normal limits as no evidence of impairment.
  19. Why would NREM2 correlate with semantic fluency only in women? No link to hormonal or neuroanatomical differences.
  20. The small sample size and lack of controls are noted but not contextualized (e.g., how this affects generalizability).
  21. Reframe cognitive differences as subtle rather than "impairment" unless clinical relevance is demonstrated.
  22. Propose mechanisms(e.g., estrogen’s role in sleep-dependent memory consolidation) for observed gender differences.
  23. What are the clinical implications: Should women with OSA receive earlier cognitive screening? Expand on this.
  24. The conclusion needs improvement as it’s too speculative (e.g., "higher susceptibility to cognitive decline") given the cross-sectional design.
  25. There’s no actionable recommendations for clinicians or researchers in the conclusion.

Round 2

Reviewer 1 Report

Comments and Suggestions for Authors

The revisions have been made quite well; however, the small sample size remains a concern. Additionally, I did not find the new title appropriate—it does not align well with the aim of the study.

Author Response

We fully recognize the limitations due to the small sample size, and we sincerely thank the reviewer for this comment, which encourages us to expand the cohort in future works.

Regarding the title “Gender Differences in Obstructive Sleep Apnea: A Preliminary Clinical and Polysomnographic Investigation”, we would like to kindly clarify that it has not been altered during the revision process. Therefore, we kept this title as it was in the first submission.

Reviewer 2 Report

Comments and Suggestions for Authors

The authors have addressed my comments well.